# Molecular Mechanisms of the Teratogenic Effects of Thalidomide

**DOI:** 10.3390/ph13050095

**Published:** 2020-05-13

**Authors:** Tomoko Asatsuma-Okumura, Takumi Ito, Hiroshi Handa

**Affiliations:** Department of Chemical Biology, Tokyo Medical University, 6-1-1, Shinjuku, Shinjuku-ku, Tokyo 160-8402, Japan; tokumura@tokyo-med.ac.jp (T.A.-O.); hhanda@tokyo-med.ac.jp (H.H.)

**Keywords:** thalidomide, cereblon, ubiquitin, lenalidomide, protein degradation, PROTACs, teratogenicity

## Abstract

Thalidomide was sold worldwide as a sedative over 60 years ago, but it was quickly withdrawn from the market due to its teratogenic effects. Thalidomide was later found to have therapeutic effects in several diseases, although the molecular mechanisms remained unclear. The discovery of cereblon (CRBN), the direct target of thalidomide, a decade ago greatly improved our understanding of its mechanism of action. Accumulating evidence has shown that CRBN functions as a substrate of Cullin RING E3 ligase (CRL4^CRBN^), whose specificity is controlled by ligands such as thalidomide. For example, lenalidomide and pomalidomide, well-known thalidomide derivatives, degrade the neosubstrates Ikaros and Aiolos, resulting in anti-proliferative effects in multiple myeloma. Recently, novel CRBN-binding drugs have been developed. However, for the safe handling of thalidomide and its derivatives, a greater understanding of the mechanisms of its adverse effects is required. The teratogenic effects of thalidomide occur in multiple tissues in the developing fetus and vary in phenotype, making it difficult to clarify this issue. Recently, several CRBN neosubstrates (e.g., SALL4 (Spalt Like Transcription Factor 4) and p63 (Tumor Protein P63)) have been identified as candidate mediators of thalidomide teratogenicity. In this review, we describe the current understanding of molecular mechanisms of thalidomide, particularly in the context of its teratogenicity.

## 1. Introduction

Thalidomide (Figure 1A) was first developed by Chemie Grünenthal (West Germany) in 1957 and was soon in use worldwide as a sedative. The use of thalidomide spread to more than 40 countries, but this drug was withdrawn from the market in 1961, as it was revealed to cause teratogenicity when taken during early pregnancy [1,2,3,4]. Clinical studies, however, demonstrated the therapeutic efficacy of thalidomide in several intractable diseases. First, in 1965, thalidomide was reported to be effective in erythema nodosum leprosum (ENL), an inflammatory complication of leprosy [5]. During the 1980s to early 1990s, thalidomide was shown to be effective in certain autoimmune diseases such as rheumatoid arthritis, Behcet’s disease, and chronic graft versus host disease [6,7,8,9]. Furthermore, in the early 1990s, thalidomide was reported to inhibit tumor necrosis factor (TNF)-alpha production and human immunodeficiency virus (HIV) replication [10,11,12]. In 1994, thalidomide was demonstrated to have anti-angiogenic activity, which suggested anti-cancer activity [13]. In 1999, thalidomide was shown to be effective against multiple myeloma, a malignant B cell lymphoma [14]. Based on these findings, in 1998 and 2006, thalidomide was approved by the Food and Drug Administration (FDA) for the treatment of ENL and multiple myeloma, respectively [15,16]. As the precise molecular mechanisms of thalidomide teratogenicity remains unclear, thalidomide prescription is strictly controlled by a program called the Thalidomide Risk Evaluation and Mitigation Strategy (REMS), formerly known as the System for Thalidomide Education and Prescribing Safety (STEPS) [17,18]. In Brazil, however, where leprosy is a common disease among the poor, the use of thalidomide led to a tragic increase in birth defects. Although the package was marked with a pictogram to prohibit its use by pregnant women, it was mistaken for a contraceptive due to poor literacy. Elucidating the molecular mechanisms of thalidomide embryopathy remains an urgent matter [19,20,21].

As the therapeutic efficacy of thalidomide was demonstrated, many thalidomide derivatives with greater potency were developed, yet the molecular mechanisms underlying the effects of thalidomide, such as inhibition of oxidative stress or angiogenesis, remained uncertain [22,23,24,25]. The most important question was to identify the direct target of thalidomide.

A decade ago, we identified cereblon (CRBN) as a primary target of thalidomide teratogenicity [26]. Since then, our understanding of the mechanisms of action of thalidomide have advanced significantly. Currently, CRBN is thought to act basically as a subunit of a ligand-dependent E3 ubiquitin ligase complex whose substrate recognition can be controlled by thalidomide or its related compounds [27]. CRBN is required for both the teratogenic effects and the therapeutic effects of thalidomide and its derivatives. Recently, CRBN-binding drugs have vividly been developed [28]. In this review, we introduce the basic functions of CRBN and discuss our current understanding of the molecular mechanisms of thalidomide, mainly focusing on its teratogenicity.

## 2. Teratogenic Activity of Thalidomide

When pregnant women took thalidomide between day 20 and day 36 after fertilization, multiple birth defects occurred [29]. A single tablet (50 mg) of thalidomide was enough to induce developmental defects [29]. A broad spectrum of birth defects was reported, including malformations of the limb, ear, eye, internal organs, face, genitalia, and central nervous system [29,30,31]. Even during the thalidomide-sensitive time period, by comparison, the earlier stages are particularly prone to more serious damages. First, the damage caused by taking thalidomide between day 20 and day 24 after fertilization appears as missing external ear. Thalidomide intake after day 24 causes multiple phenotypes such as damage in the inner ear, ear deformation, ocular anomalies, and upper limb damage (phocomelia, amelia), or hip dislocation. Damage to the lower limbs is seen in the comparatively late intake of the drug during the thalidomide sensitive time window, which is after day 27. Malformations in thumbs are seen by taking thalidomide from day 24 and even after day 31 [29]. The mortality rate for infants with thalidomide-induced birth defects was reported to be 30%–40%. Thalidomide caused imperforate anus and other gastrointestinal deformities in many infants, contributing to early death [31,32]. In addition, an unknown number of miscarriages were caused by thalidomide.

Limb defects were very frequently observed [31]. Both upper limbs and lower limbs were affected by thalidomide. Two types of limb defects are induced by thalidomide, phocomelia and amelia. The limb is composed of the stylopod, zeugopod, and autopod. Phocomelia describes an abnormal limb with a stylopod, a truncated or absent zeugopod, and a nearly intact autopod, while amelia is a complete loss of the limb [29,33,34]. Polydactyly was also observed in deformed limbs, including phocomelia [1,29,35,36]. Defects of the shoulder and hip joint points were reported [34]. Auricular defects were also very frequently observed, including anotia, mild malformation of the external ear, and hearing loss [29,31,35]. Ocular anomalies included uveal coloboma, glaucoma, and microphthalmia. With respect to the internal organs, kidney malformations, heart defects, and structural chest defects were frequently observed [34,37,38]. Facial palsy and facial asymmetry were also common [31,35]. Thalidomide was also reported to potentially affect facial muscles and facial nerves [29,31,35]. Autism and intellectual disability were also reported [31,35,39].

Thalidomide causes limb defects in humans, monkeys, rabbits, chicks, and zebrafish [27,30,33]. In monkeys and rabbits, both amelia and phocomelia occur. In chicks, only amelia occurs [33]. In zebrafish, thalidomide inhibits the development of the pectoral fins along the proximodistal axis [26,40,41]. Although the fin is structurally different from the limb in mammals and chicks, the molecular pathways are evolutionary conserved. Thalidomide inhibits chondrogenic differentiation in pectoral fins. Pectoral fins are composed of the endoskeletal disc, the scapulocoracoid, and the cleithrum. Thalidomide treatment in the early stage of development resulted in severe defects in chondrogenesis and retardation of the endoskeletal disc and cleithrum [40]. Rodents are resistant to limb deformities induced by thalidomide. It was reported that thalidomide did not induce limb defects in rats even at doses of up to 4000 mg/kg [42]. Why mice and rats are resistant to thalidomide teratogenicity remains unknown.

In vertebrates, fibroblast growth factor 8 (FGF8) is essential for the development of limbs, including fins [43,44]. FGF8 is expressed in the apical ectodermal ridge (AER), the distalmost end of the developing limbs. Thalidomide was shown to reduce the expression of FGF8 in the AER in rabbits, chicks and zebrafish [26,45]. Downregulation of FGF8 leads to induction of pro-apoptotic genes and therefore to malformation of the limbs. Therefore, the effects of thalidomide are likely mediated through evolutionarily common signal transduction pathways in different vertebrates.

## 3. The Direct Target of Thalidomide

In 2010, the identification of CRBN as a thalidomide-binding protein represented a major advance in understanding the molecular mechanism of thalidomide [26]. We immobilized various small chemical bioactive compounds onto ferrite glycidyl methacrylate (FG) beads to study their targets [46,47,48]. Affinity purification using thalidomide-immobilized FG beads led to the identification of CRBN and DNA damage-binding protein 1 (DDB1) as thalidomide-binding proteins. The function of CRBN was unknown at the time, although it was thought to be related to mental retardation and intellectual disability [49]. A clue to the function of CRBN was DDB1, the protein co-purified with CRBN. DDB1 forms a complex with Cullin RING type E3 ubiquitin ligase (CRL4) [50,51,52,53]. CRBN forms a complex with Cullin 4 (Cul4), DDB1, and regulator of Cullins-1 (Roc1) and functions as a substrate of this CRL4 complex (CRL4^CRBN^). The autoubiquitination of CRBN was shown to be inhibited by thalidomide. It was found that the CRBN^Y384A/W386A(YW/AA)^ mutant did not bind to thalidomide. The zebrafish and chick developmental model systems were utilized to evaluate whether CRBN was genuinely involved in mediating thalidomide-induced teratogenicity. When CRBN^YW/AA^ was overexpressed, the teratogenic phenotypes of thalidomide were reversed in both chicks and zebrafish. The expression of FGF8 was restored by CRBN^YW/AA^ expression even after thalidomide treatment. These findings demonstrated that CRBN was a primary target of thalidomide and critically involved in thalidomide teratogenicity.

## 4. CRBN as a Therapeutic Target of Thalidomide and Its Derivatives

The finding that CRBN was a critical, direct target of thalidomide and functioned as a subunit of a CRL4 E3 ligase greatly advanced our understanding of the molecular mechanism of thalidomide and its derivatives. The therapeutic effects of thalidomide and its derivatives, rather than the teratogenic effects, became the focus of further research. As mentioned previously, the remedial effect of thalidomide against multiple myeloma led to the development of its derivatives lenalidomide and pomalidomide—both of which are now approved by the FDA. Pomalidomide is a compound in which an amino group is added to the phthalimide of thalidomide, and lenalidomide has the structure of pomalidomide without the carbonyl group on the phthalimide moiety (Figure 1B,C). Both compounds are called immunomodulatory drugs (IMiDs) and are reported to have more potent immunomodulatory activity than thalidomide [15,54].

An intriguing question was whether CRBN was involved in the therapeutic effects of lenalidomide and pomalidomide. In 2011, Stewart and colleagues found that knockdown of CRBN by RNA interference (RNAi) blocked the inhibition of cell proliferation by lenalidomide or pomalidomide in several multiple myeloma cell lines [55]. The group also reported that the expression of CRBN was considerably lower in pomalidomide-resistant cell lines [55]. Celgene Corporation and our group confirmed their data in 2012. We also demonstrated that compounds containing a glutarimide moiety, such as lenalidomide and pomalidomide, bind to CRBN [56], and lenalidomide and pomalidomide bound to CRBN more strongly than thalidomide. These results suggested that CRBN is required not only for the teratogenic effects but also the therapeutic effects of thalidomide and its derivatives.

## 5. Ligand-Dependent Substrate Recognition of CRL4^CRBN^

Since CRBN was shown to be involved in the anti-cancer effects of thalidomide and its derivatives, researchers next investigated the relevant CRBN substrates. Since then, several CRBN substrates have been identified.

### 5.1. Ikaros and Aiolos

In 2014, two independent groups found lenalidomide-dependent CRL4^CRBN^ substrates, Ikaros (IKZF1) and Aiolos (IKZF3), in multiple myeloma cell lines [57,58]. Ikaros and Aiolos belong to the Ikaros zinc finger family (IKZF) [59]. In the presence of lenalidomide, Ikaros and Aiolos were polyubiquitinated by CRL4^CRBN^ and subsequently degraded in the proteasome. Such ligand-dependent substrates are called neosubstrates. Other IKZF family members, IKZF2 and IKZF4, were not degraded by lenalidomide. The 146th amino acid in Ikaros and 147th in Aiolos is glutamine (Q146 in Ikaros and Q147 in Aiolos), which are replaced with histidine in IKZF2 and IKZF4, respectively. Neither Ikaros^Q146H^ nor Aiolos^Q147H^ was degraded by lenalidomide, while IKZF4^H188Q^ was degraded. In addition, myeloma cell lines expressing Ikaros^Q146H^ or Aiolos^Q147H^ were resistant to lenalidomide. It was concluded that the anti-myeloma effect of lenalidomide is primarily due to the degradation of Ikaros and Aiolos. Later, Celgene and our group showed that not only lenalidomide but also pomalidomide induced the degradation of Ikaros and Aiolos, resulting in upregulation of interleukin (IL)-2 in T cells [60].

### 5.2. CK1α

Lenalidomide is the only IMiD that is approved for the treatment of myelodysplastic syndrome (MDS) with deletion of chromosome 5q (5q-) [61,62]. However, the mechanism was unclear. In 2015, in addition to Ikaros and Aiolos, casein kinase alpha (CK1α) was identified as a lenalidomide-dependent CRL4^CRBN^ neosubstrate [63]. The 5q- MDS cells carry a deletion in the chromosome region containing the CSNK1A1 gene, resulting in haploinsufficient expression of CK1α. In such 5q- cells, the degradation of CK1α by lenalidomide resulted in cell death. Furthermore, this degradation of CK1α by lenalidomide was considerably weaker than that induced by thalidomide or pomalidomide. This finding suggested that the neosubstrates recognized by CRL4^CRBN^ differ depending on the ligand.

### 5.3. GSPT1

Celgene has been developing numerous thalidomide derivatives. Among them, CC-885 was shown to possess potent anti-proliferative activity against various cancer cell lines and a noteworthy effect against acute myelogenous leukemia (AML) and AML-derived cell lines. Notably, anti-AML activity was not found in thalidomide or any previously characterized IMiDs. The structure of CC-885 is similar to that of lenalidomide, with an extended structure (a urea and a chloro-methyl-phenyl group) (Figure 1D). Our group performed immuno-affinity purification of the CRBN–CC-885 complex and identified a CC-885-dependent neosubstrate, G1-to-S phase transition 1 (GSPT1), also known as eukaryotic peptide chain release factor GTP-binding subunit ERF3A [64]. Biochemical studies have shown that CC-885 induces anti-AML effects via the degradation of GSPT1. Currently, several derivatives of CC-885 have been developed [65]. CC-90009 (Figure 1E) is being tested in clinical trials [66].

### 5.4. ZFP91 and Other Zinc Finger Proteins

The search for CRL4^CRBN^ neosubstrates has continued and, in 2017, another zinc finger motif-containing neosubstrate was found in the non-hematological cell lines HCT116 (colon cancer) and 293T [67]. This finding suggested the hypothesis that at least a fraction of CRL4^CRBN^ neosubstrates share an IKZF-like zinc finger. To further investigate the zinc finger hypothesis, a C2H2 zinc finger library was screened via proteome-wide mass spectrometry. This study identified several new CRL4^CRBN^ neosubstrates, including ZNF692 (Zinc Finger Protein 692) [68].

## 6. Structure of the CRBN–Drug–Neosubstrate Complex

The structure of the direct target of thalidomide has been intriguing to structural biologists. In 2014, Thoma, Fischer, and colleagues reported the X-ray structure of a chimeric complex of human DDB1 and chick CRBN bound to thalidomide [69]. Celgene and our group also reported the X-ray structure of human CRBN and human DDB1 bound to lenalidomide [70]. Chick CRBN is highly homologous to human CRBN. Chick CRBN is composed of at least three domains, a seven-stranded β-sheet located in the amino-terminal domain (NTD, residues 1–185), an α-helix bundle domain (HBD, residues 186–317) containing seven helices, and a carboxy-terminal domain (CTD, residues 318–445) composed of eight β-sheets. The NTD and the HBD are related to the N-terminus of Lon protease. The structure of DDB1 is composed of three β-propeller blades (BPA, BPB, and BPC). The HBD of CRBN binds BPA and BPC. The CTD contains a zinc finger domain and the thalidomide-binding domain (TBD). The zinc finger consists of C323, C326, C391, and C394 in humans. The physiological significance of the zinc finger is still unknown, but it is noteworthy that the C391R mutation of CRBN was reported to be associated with intellectual disability [71]. TBD possesses three tryptophans (W380, W386, and W400) that form a pocket called the tri-trp pocket. The glutarimide moiety of thalidomide and its derivatives can be inserted into the pocket.

Subsequent reports by these same groups in 2016 showed that the ligands form a molecular glue between CRBN and its neosubstrates. The Thoma group reported the X-ray structure of the CK1α–lenalidomide–CRBN/DDB1 complex and Celgene and our group reported the structure of the GSPT1–CC-885–CRBN/DDB1 complex [64,72]. Both CK1α and GSPT1 contain a β-hairpin motif, which is essential for binding to CRBN. In the CK1α complex, lenalidomide acts as a molecular glue between CRBN and CK1α. The G40 residue of CK1α in the hairpin motif is sandwiched between phthalimide and CRBN. The glycine is critical for the interaction with lenalidomide. One of the carbonyl groups in the phthalimide moiety prevents steric clash. Therefore, lenalidomide degrades CK1α more efficiently than thalidomide or pomalidomide. In the GSPT1 complex, CC-885 connects CRBN to GSPT1. The chloro-methyl-phenyl group of CC-885 is important for binding to GSPT1. The G575 of GSPT1, like the G40 of CK1α, is essential for its interactions with CC-885 and CRBN. The F150 residue of CRBN, which is in the NTD, is important for binding to GSPT1, but is not essential for interactions with CK1α and Ikaros. Although the functions of the NTD of CRBN are still largely unknown, the NTD is required for its interactions with several neosubstrates. In 2018, the structure of CRBN bound to the zinc finger motif of Ikaros or the related neosubstrate ZNF692 in the presence of pomalidomide was determined, and the amino group of pomalidomide was shown to contribute to the interaction with Ikaros/ZNF692 and CRBN [68].

Thalidomide and its derivatives have one chiral center. Thalidomide and other IMiDs are usually used as a mixture of the (S)-isomer and the (R)-isomer. Previously, researchers thought that thalidomide exerts different effects depending on the enantiomer [73]. However, the isomers rapidly racemize under physiological conditions [74]. It therefore remained unclear whether thalidomide acted differently on different optical isomers. Researchers have shown that the (S)-enantiomer bound approximately 10-fold more strongly to CRBN [70,75] than the (R)-enantiomer. The crystal structures of CRBN and each enantiomer revealed that both enantiomers bind the tri-trp pocket. However, the bound form of (S)-enantiomer to CRBN exhibited a more relaxed conformation of its glutarimide ring [69,70,75]. The conformation of the (R)-enantiomer is twisted to avoid steric clashes, resulting in a weaker binding activity. The (S)-enantiomer more effectively decreased Ikaros protein in multiple myeloma and induced fin defects in zebrafish [75]. Although the (R)-enantiomer possesses weak CRBN-binding activity, it might be a supplier of (S)-enantiomer by racemization under physiological conditions.

Through structural biological analyses, species-specific effects of thalidomide and its derivatives have been partly elucidated. In rodents, neither teratogenic effects nor therapeutic effects of thalidomide and its derivatives have been observed [76,77]. Mouse CRBN is 95% homologous to human CRBN and can bind to thalidomide. However, it was shown that lenalidomide and pomalidomide did not degrade neosubstrates such as Ikaros or CK1α in mouse cells or human cells in which CRBN was replaced with mouse CRBN [63]. There are two critical amino acids in human CRBN, E377 and V388. In rodents, these amino acids are substituted to valine and isoleucine, respectively. The V388I substitution of CRBN abolishes the interaction with Ikaros and CK1α in the presence of thalidomide or IMiDs [63]. Structural studies have shown that CRBN binds to Ikaros and CK1α at V388 [68,72]. The V388I mutation induces steric clash and therefore prevents binding. The E377V substitution abolished the interaction with GSPT1 in the presence of CC-885 because E377 binds to the urea of CC-885 [62]. Lenalidomide has been shown to degrade mouse CK1α in “humanized” mouse cells expressing mouse CRBN^I391V^ [63]. Researchers constructed humanized mice expressing mouse CRBN^I391V^ [78,79] and found that lenalidomide acted on the hematopoietic stem cells in humanized mice. Thalidomide induced fetal loss in these mice, but limb defects were not observed [79]. Thalidomide resistance in rodents therefore remains a mystery.

## 7. Teratogenic Mechanisms Associated With CRBN

While the substrates of thalidomide and its derivatives and their roles in its therapeutic effects have been elucidated, the molecular basis of the teratogenic effects remained unclear. Four downstream factors have been identified as candidate players in thalidomide—CRBN-mediated teratogenicity.

### 7.1. MEIS2

Fischer and colleagues first performed a structural analysis of DDB1–CRBN bound to thalidomide or its derivatives and evaluated the binding of the known IKZF family neosubstrates. They examined the ubiquitination of approximately 9000 proteins by CRL4^CRBN^ using human protein microarrays and identified MEIS2 as a protein that was ubiquitinated by CRBN and was stabilized upon treatment with thalidomide or other IMiDs [69]. MEIS2 had been hypothesized to negatively regulate limb outgrowth [80], which made it an intriguing downstream target of thalidomide. However, whether the accumulation of MEIS2 in response to thalidomide treatment is involved in limb malformation has not been shown to be investigated using any suitable animal model.

### 7.2. CD147

The CD147 (Cluster of Differentiation 147) /MCT1 (Monocarboxylate Transporter 1) complex was identified as a CRBN-binding partner by tandem affinity purification [81]. Lenalidomide acted as a competitive inhibitor of the CRBN–CD147/MCT1 interaction. Inhibition of this interaction destabilized CD147/MCT1, independent of the ubiquitination of CRL4^CRBN^ [81]. CD147 is expressed on the surface of multiple myeloma cells and functions as a receptor for secreted cyclophilin A, which promotes aggregation and homing to the bone marrow [82]. Therefore, destabilization of the CD147/MCT1 complex by lenalidomide attenuates cell proliferation. Interestingly, 5q- MDS has elevated expression of CD147 compared with non5q- MDS, which may provide an explanation for the observation that lenalidomide is ineffective in 5q- MDS [81]. In addition to its clinical role, the CD147/MCT1 complex was investigated in the context of thalidomide-induced teratogenicity. Zebrafish in which CD147 had been knocked down showed teratogenic phenotypes such as malformation of the head, pectoral fins, and eyes [81]. Reduced expression of FGF8 was demonstrated in the fin of CD147 knockdown zebrafish [81].

### 7.3. SALL4

In 2018, two independent groups reported a C2H2 zinc finger transcription factor SALL4 (Spalt Like Transcription Factor 4) as a thalidomide-dependent neosubstrate of CRL4^CRBN^. First, Fischer’s group identified neosubstrates that decreased upon thalidomide, lenalidomide, or pomalidomide treatment by mass spectrometry using human embryonic stem cells (hESC), as no animal model was similar enough to mimic thalidomide syndrome in humans [83]. SALL4 was selected from the list of proteins degraded upon thalidomide treatment because it was previously identified as the causal gene of hereditary diseases such as Duane Radial Ray syndrome, Okihiro syndrome and Holt–Oram syndrome [84,85,86]. These syndromes partly overlap with thalidomide embryopathy [87]. In addition, Chamberlain’s group independently identified SALL4 on the basis of its structural similarity with the known zinc finger-type neosubstrates [88]. Both groups showed that degradation of human SALL4 by human CRBN was thalidomide dependent, and the crucial glycine identified for the degradation was G416. In addition, it was shown that mouse SALL4 was not degraded. Chamberlain’s group observed SALL4 degradation in the rabbit fetus, another model animal for thalidomide teratogenicity [88]. Both studies concluded that SALL4 is a neosubstrate responsible for the teratogenic effects of thalidomide.

A follow-up study reported the effects of thalidomide and IMiDs against differentiating human induced pluripotent stem cells (hiPSCs). The authors demonstrated that thalidomide-dependent SALL4 degradation in hiPSCs was abolished by a mutation in CRBN of valine to isoleucine at position V388. Similar effects were observed with SALL4 G416A. These mutations desensitized hiPSCs to the effects of thalidomide or IMiDs upon differentiation to lateral plate mesodermal (LPM)-like cells. The mutated hiPSCs retained sensitivity to inhibition of differentiation by all-trans retinoic acid (atRA) or SB431542, which are other known teratogens. This study suggests that SALL4 may be a crucial neosubstrate involved in limb malformation [89].

The role of SALL4 has been studied in a mouse model. SALL4 knockout mice were embryonic lethal, and in heterozygotes, the phenotype varied; some litters showed phenotypic changes in the heart (interventricular septum loss) and did not survive long, yet those that survived longer than 3 weeks showed no obvious phenotypes of Okihiro syndrome, such as abnormalities in the digits [90]. When SALL4 heterozygotes were crossed with heterozygotes of SALL1, another SALL family member, the pups showed Okihiro syndrome-like phenotypes in the anorectal system, heart, brain, and/or kidneys, and did not survive long. Further, homozygous SALL4 knockout was also lethal in zebrafish, which showed morphological abnormalities in the heart and eyes and had shortened bodies, although the embryos possessed pectoral fins [91].

Indeed, SALL4 had already been proposed to be a downstream target of thalidomide before CRBN was found; Knobloch and Ruther showed in 2008 that SALL4 mRNA decreased upon thalidomide treatment in chickens [33]. This pathway seemingly differs from the thalidomide–CRL4^CRBN^ protein degradation axis, and there may be more to be determined about the relationship between thalidomide and SALL4.

### 7.4. p63

Guerrini and colleagues have been studying the TP63 (Tumor Protein P63 or p63) gene [92,93]. TP63 is the causal gene of genetic syndromes with multiple birth defects. Congenital limb malformations, ectodermal dysplasias, and facial clefts are the main characteristics of human patients with TP63 mutations such as ectodermal dysplasia and cleft lip/palate (EEC) syndrome and acro-dermato-ungual-lacrimal-tooth (ADULT) syndrome [94]. In mice, p63 knockout causes abnormalities in the development of epithelial structures, including limbs, and the fins are abolished in zebrafish with p63 knocked down [95,96,97]. Guerrini hypothesized p63 as among the downstream targets of thalidomide. Our group worked together with the Guerrini group on these studies, and our collaborative group identified a relationship between p63 and CRBN. More than ten isomers of p63 have been identified, depending on promoter use and splice variation [98]. We evaluated the major isoforms, ΔNp63α and TAp63α, and found that both isoforms were thalidomide-dependent neosubstrates of CRL4^CRBN^ [40]. Although p63 is a non-C2H2 zinc finger-type neosubstrate, we identified a glycine that was important for its degradation, and mutant versions of ΔNp63α and TAp63α were not degraded by thalidomide treatment. Furthermore, zebrafish were used to examine the thalidomide teratogenic effect in animals. ΔNp63 is expressed in the AER and epithelial tissue, while TAp63 is mainly expressed in the heart and the ear [96,99,100]. As mentioned previously, thalidomide-treated zebrafish have abnormalities in the pectoral fins and otic vesicles. When ΔNp63 or TAp63 was knocked down, zebrafish showed defects in the pectoral fins or otic vesicles, respectively. When the non-degraded mutants of ΔNp63 or TAp63 were overexpressed, the abnormal development of the fins or otic vesicles was reversed in thalidomide-treated zebrafish. In addition to these phenotypic observations, the expression of downstream targets was examined in each tissue (FGF8, a crucial regulator of limb/fin development, and Atoh1, an essential transcription factor for the development of sensory neurons and cochlea development) [100]. The expression of both targets was downregulated upon thalidomide exposure or ΔNp63/TAp63 knockdown, respectively. These findings were confirmed with overexpression of non-degraded ΔNp63/TAp63 mutants. Taken together, we concluded that at least in zebrafish, ΔNp63 and TAp63 were thalidomide-dependent CRL4^CRBN^ neosubstrates responsible for teratogenicity. Interestingly, there are a few reports demonstrating the protective role of ΔNp63α against oxidative stress. In these papers, ΔNp63α was shown to confer resistance to oxidative stress-induced cell death [101,102]. In addition to FGF8 downregulation, the thalidomide-dependent breakdown of ΔNp63α may increase oxidative stress, which generally fits with the previously mentioned oxidative stress hypothesis. Our study in zebrafish was limited to defects of the limbs and otic vesicles, and thalidomide-dependent malformations in other tissues have not been examined. We cannot exclude the possibility that there are additional CRBN neosubstrates associated with other defects of thalidomide.

With all of these findings taken together, we have now come to understand some part of the mechanism of thalidomide embryopathy; that is, thalidomide binds to its only target, CRBN, which affects various downstream pathways, resulting in the accumulation of MEIS2, the destabilization of CD147/MCT, and the breakdown of multiple neosubstrates such as SALL4, ΔNp63, TAp63, and likely others yet to be identified (Figure 2). These changes potentially lead to varied birth defects.

## 8. Concluding Remarks

This decade saw remarkable progress in our understanding of the targets and underlying molecular mechanisms of thalidomide. First, thalidomide binds to CRBN, which recruits a neosubstrate, then ubiquitinates the bound neosubstrate. Ubiquitinated neosubstrates are degraded via the ubiquitin–proteasome pathway, and various effects occur, depending on the neosubstrate.

More thalidomide derivatives, now called cereblon E3 ligase modulators (CELMoDs), have been developed and are now being tested in clinical trials [66,103]. CC-122 (avadomide) (Figure 1F) has broader effects than IMiDs and is effective against diffuse large B cell lymphomas (DLBCLs) and solid tumors [104,105]. CC-220 (iberdomide) (Figure 1G) has stronger Aiolos/Ikalos degradation activity than IMiDs and is being studied for the treatment of systemic lupus erythematosus (SLE) [106,107,108]. Among the most recently developed CELMoDs is CC-92480 (Figure 1H), which is being studied for the treatment of lenalidomide-resistant multiple myeloma [109].

Furthermore, new approaches utilizing the drug-dependent CRL4^CRBN^ ubiquitination activity are being established. By combining thalidomide or other CRBN-binding compounds with other low-molecular-weight compounds that interact with pathogenic proteins, CRL4^CRBN^ can be recruited to degrade specific proteins of interest (POIs). This technology is called Proteolysis-Targeting Chimeras (PROTACs) and was originally proposed by Crews and Deshaies [110]. To achieve this targeted protein degradation approach, many PROTAC molecules are being synthesized and tested for clinical use [111,112,113,114,115]. For example, dBET1 (Figure 1I), among the earliest CRBN-based PROTACs to be developed by Bradner and colleagues, is a fusion of thalidomide and JQ1 [113]. JQ1 is an acetylated-lysine-like compound that binds to bromodomain and extra-terminal (BET) motif-containing transcription factors such as BRD4 (Bromodomain containing 4) and exerts its effects by competitively inhibiting the binding of these transcription factors to acetylated histones. As silencing of BRD4 downregulates the expression of the MYC (v-Myc Myelocytomatosis Viral Oncogene Homolog) oncogene, JQ1 is a potent anti-cancer molecule [116]. Nevertheless, dBET1 has improved anti-cancer activity, as it degrades the transcription factor itself directly, which leads to comparatively rapid inhibition.

Although expectations are high for the clinical use of CRBN-binding compounds as a means of targeted protein degradation, the potential for serious adverse effects needs to be taken into consideration. Downstream targets of thalidomide are gradually being identified, but we have a long way to go to completely elucidate the mechanism of teratogenicity. The structural comparison of the therapeutic neosubstrates with the teratogenic neosubstrates might contribute to develop new safer CRBN-binding drugs [117].

Finally, the function of CRBN in the absence of thalidomide or its derivatives remains mostly uncharacterized. As briefly mentioned above, CRBN was originally reported to be related to intellectual disability, and it was also reported that CRBN interacts with the BK_Ca_ channel α subunit (Slo) (BK_Ca_ stands for Large-conductance Ca^2+^- and voltage-gated big K^+^) [118,119]. The function of CRBN without thalidomide or other drugs has been examined in animals. Forebrain-specific conditional CRBN knockout mice displayed learning disabilities, and knockdown of CRBN in zebrafish impaired brain development [120,121]. These studies suggest that CRBN function is related to the developing brain, but the function of ligand-unbound CRBN remains unknown. Uridine was also reported to bind to CRBN, and a high concentration of uridine mimicked the thalidomide fin deformity in zebrafish [122]. There might be additional natural ligands or metabolites that bind to CRBN [123]. Further understanding of the basic function of CRBN may lead to the discovery of new biological phenomena and will contribute to the development of safer and more effective drugs.

## Figures and Tables

**Figure 1 pharmaceuticals-13-00095-f001:**
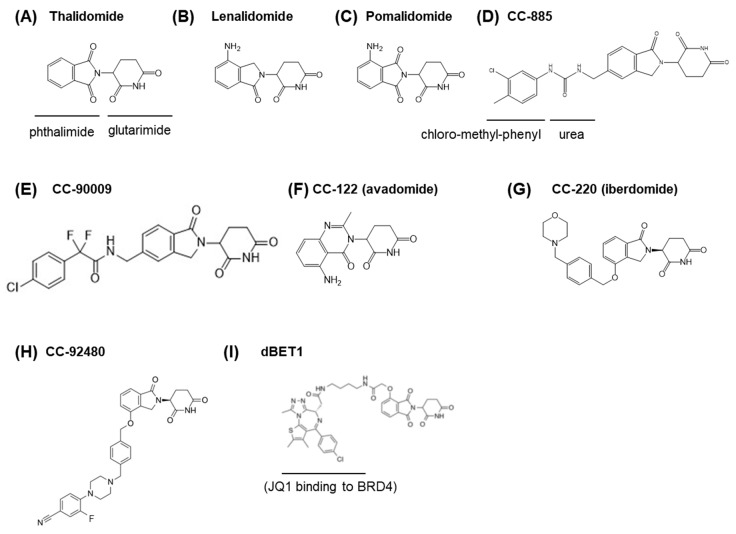
Structure of CRBN-binding drugs. (**A**) Thalidomide. (**B**) Lenalidomide. (**C**) Pomalidomide. (**D**) CC-885. (**E**) CC-90009 (**F**) CC-122. (**G**) CC-220. (**H**) CC-92480. (**I**) dBET1, composed of JQ1 (a BRD4 inhibitor) and thalidomide.

**Figure 2 pharmaceuticals-13-00095-f002:**
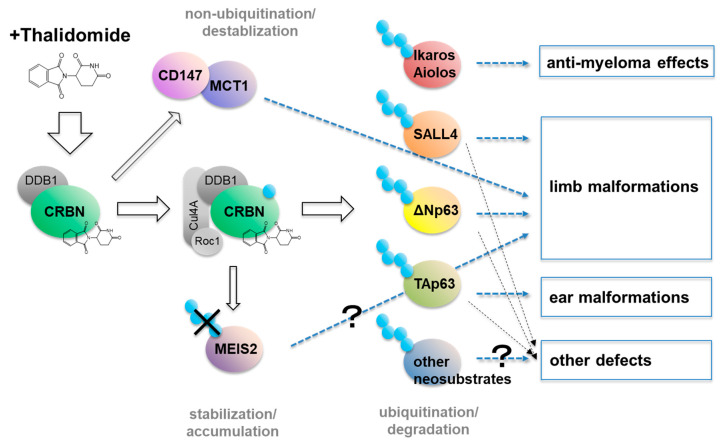
Model of the molecular mechanism of thalidomide. Thalidomide exerts multiple functions after binding to CRBN: (i) the non-ubiquitination process of destabilizing the CD147/MCT1 complex, (ii) the blockade of ubiquitination that stabilizes MEIS2, and (iii) the ubiquitin-dependent degradation of several neosubstrates including SALL4 and p63. These multiple downstream targets of the thalidomide–CRBN axis result in the various effects of thalidomide.

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
