# Peer review of "Molecular Mechanisms of the Teratogenic Effects of Thalidomide"

_pharmaceuticals, 2020, doi:10.3390/ph13050095_

Round 1

Reviewer 1 Report

Molecular Mechanisms of the Teratogenic Effects of Thalidomide authored by Tomoko Asatsuma-Okumura , Takumi Ito,  and Hiroshi Handa is an exceptional review of the compound thalidomide and the molecular substrates of thalidomide that mediate its teratogenicity.

 The thalidomide tragedy occurred in the late 1950s and early 1960s when pregnant women were given a drug known as thalidomide to alleviate morning sickness. What resulted was tragic in that approximately 10,000-20,000 babies worldwide were born with birth defects typified by abnormal limb development. It took about 50 years to identify thalidomide's target which is known as cereblon. The group writing this current review were the ones who identified cereblon about a decade ago. The thalidomide story is truly a fascinating one that is rooted in both teratogenicity and therapeutics. This review primarily focuses on the teratogenic influences of thalidomide at the molecular levels.

The review is exceptionally well written and well cited and this reviewer only has a couple of very minor concerns that the authors may want to address, but these clearly are not mandatory. There is also one sentence in the manuscript that may need to be rewritten. It is presented below.

 1) As it relates to the thalidomide tragedy itself, the authors may wish to point out that the different birth defects that occurred were linked to various critical windows of development within the general window of exposure which was 20-36 days after fertilization. This is something that is easy to research. For example, if thalidomide was taken very early in pregnancy, ears were effected; a little later upper arms were impacted, etc.

2) This reviewer is highly appreciative that the authors acknowledged the issue of thalidomide use in Brazil. Although the paper referenced for this tragedy was 2007, this remains a major Public Health issue in Brazil. Present day Brazil may be the only country in the world that still has kids who have suffered from thalidomide teratogenicity. Possibly consider adding this reference:

Sales Luiz Vianna F, de Oliveira MZ, Sanseverino MT, Morelo EF, de Lyra Rabello Neto D, Lopez-Camelo J, Camey SA, Schuler-Faccini L. Pharmacoepidemiology and thalidomide embryopathy surveillance in Brazil. Reprod Toxicol. 2015

3) Original Sentence: Lines 117-118 To determine whether CRBN is genuinely involved in thalidomide teratogenicity, chicken or zebrafish were used as model animals.

Suggested sentence: The zebrafish and chick developmental model systems were utilized to evaluate if CRBN was genuinely involved in mediating thalidomide-induced teratogenicity.

Lastly, please see references 53 (line 532) and 80 (line 610) in reference list: yellow highlights should be removed.

Author Response

Molecular Mechanisms of the Teratogenic Effects of Thalidomide authored by Tomoko Asatsuma-Okumura , Takumi Ito, and Hiroshi Handa is an exceptional review of the compound thalidomide and the molecular substrates of thalidomide that mediate its teratogenicity.
The thalidomide tragedy occurred in the late 1950s and early 1960s when pregnant women were given a drug known as thalidomide to alleviate morning sickness. What resulted was tragic in that approximately 10,000-20,000 babies worldwide were born with birth defects typified by abnormal limb development. It took about 50 years to identify thalidomide's target which is known as cereblon. The group writing this current review were the ones who identified cereblon about a decade ago. The thalidomide story is truly a fascinating one that is rooted in both teratogenicity and therapeutics. This review primarily focuses on the teratogenic influences of thalidomide at the molecular levels.
The review is exceptionally well written and well cited and this reviewer only has a couple of very minor concerns that the authors may want to address, but these clearly are not mandatory. There is also one sentence in the manuscript that may need to be rewritten. It is presented below.

1) As it relates to the thalidomide tragedy itself, the authors may wish to point out that the different birth defects that occurred were linked to various critical windows of development within the general window of exposure which was 20-36 days after fertilization. This is something that is easy to research. For example, if thalidomide was taken very early in pregnancy, ears were effected; a little later upper arms were impacted, etc.

OUR RESPONSE: We appreciate the reviewer’s comment and have added the following description in the revised manuscript.

Even during the thalidomide-sensitive time period, by comparison, the earlier stages are particularly prone to more serious damages. First, the damage caused by taking thalidomide between day 20 and day 24 after fertilization appears as the missing of the external ear. The thalidomide intake after day 24 causes multiple phenotypes such the damage in the inner ear, ear deformation, ocular anomalies’, as well as upper limb damage (phocomelia, amelia), or hip dislocation. The damages of the lower limbs are seen in the comparatively late intake of the drug during the thalidomide sensitive time window, which is after day 27. The malformations of the thumb are seen by the intake of thalidomide from 24 and even after day31 [29].

2) This reviewer is highly appreciative that the authors acknowledged the issue of thalidomide use in Brazil. Although the paper referenced for this tragedy was 2007, this remains a major Public Health issue in Brazil. Present day Brazil may be the only country in the world that still has kids who have suffered from thalidomide teratogenicity. Possibly consider adding this reference:
Sales Luiz Vianna F, de Oliveira MZ, Sanseverino MT, Morelo EF, de Lyra Rabello Neto D, Lopez-Camelo J, Camey SA, Schuler-Faccini L. Pharmacoepidemiology and thalidomide embryopathy surveillance in Brazil. Reprod Toxicol. 2015

OUR RESPONSE: We appreciate the reviewer for the comment and have added the reference (Ref-21) in the revised manuscript.

3) Original Sentence: Lines 117-118 To determine whether CRBN is genuinely involved in thalidomide teratogenicity, chicken or zebrafish were used as model animals.
Suggested sentence: The zebrafish and chick developmental model systems were utilized to evaluate if CRBN was genuinely involved in mediating thalidomide-induced teratogenicity.

OUR RESPONSE: We appreciate the reviewer for the comment. The revised manuscript has been edited accordingly.

Lastly, please see references 53 (line 532) and 80 (line 610) in reference list: yellow highlights should be removed.

OUR RESPONSE: The revised manuscript has been edited accordingly.

Reviewer 2 Report

Dear Authors,

The manuscript is well organized and written, although I would recommend using more conversational style of writing. 

The authors detailly descried the molecular mechanisms underlaying both therapeutic and teratogenic activity of thalidomide. The role of Cereblon (CRBN) as the molecular target for thalidomide has been discussed with a focus on various downstream pathways involved in teratogenesis like the accumulation of Myeloid Ecotropic Insertion Site 2 (MEIS2), the destabilization of CD147/MCT, and the breakdown of multiple neosubstrates such as SALL4, ΔNp63 and TAp63.

The presented manuscript is worth publishing.

Author Response

The manuscript is well organized and written, although I would recommend using more conversational style of writing.
The authors detailly descried the molecular mechanisms underlaying both therapeutic and teratogenic activity of thalidomide. The role of Cereblon (CRBN) as the molecular target for thalidomide has been discussed with a focus on various downstream pathways involved in teratogenesis like the accumulation of Myeloid Ecotropic Insertion Site 2 (MEIS2), the destabilization of CD147/MCT, and the breakdown of multiple neosubstrates such as SALL4, ΔNp63 and TAp63.
The presented manuscript is worth publishing.

OUR RESPONSE:
We appreciate the reviewer’s comment and would like to edit the revised text as directed by the editorial office, if necessary.